# Hemisphere-specific spatial representation by hippocampal granule cells

**Thibault Cholvin** [1] ✉ **& Marlene Bartos** [1] ✉

The dentate gyrus (DG) output plays a key role in the emergence of spatial and contextual map representation within the hippocampus during learning. Differences in neuronal network activity have been observed between left and right CA1-3 areas, implying lateralization in spatial coding properties. Whether bilateral differences of DG granule cell (GC) assemblies encoding spatial and contextual information exist remains largely unexplored. Here, we employed two-photon calcium imaging of the left or the right DG to record the activity of GC populations over five consecutive days in head-fixed mice navigating through familiar and novel virtual environments. Imaging revealed similar mean GC activity on both sides. However, spatial tuning, context-selectivity and run-to-run place field reliability was markedly higher for DG place cells in the left than the right hemisphere. Moreover, the proportion of GCs reconfiguring their place fields between contexts was greater in the left DG. Thus, our data suggest that contextual information is differentially processed by GC populations depending on the hemisphere, with higher context discrimination in the left but a bias towards generalization in the right DG.

A fundamental and fascinating feature of the mammalian brain is its capacity to acquire, store, and recall novel conscious memories. The mammalian hippocampus is particularly decisive for episodic memory. By routing spatial and contextual information through its canonical trisynaptic circuitry from the dentate gyrus (DG) to CA3 and via Schaffer collaterals to CA1, with the latter thus acting as the output node of the hippocampus, it allows the processing and transfer of contextual information to neocortical areas. Each of the hippocampal regions is characterized by specialized cellular and neuronal network operations[1–6]. Computational studies proposed that the DG circuitry is particularly crucial for the implementation of pattern separation, i.e., the ability to segregate and store similar but discrete contexts and events by activity patterns of minimally overlapping granule cell (GC) populations[4,6,7]. The DGs' segregating function is supported by the absence of direct synaptic connections among GCs[8,9] and their sparse discharges[10,11]. However, experimental data yielded conflicting results, reporting either weaker[3,12,13] or stronger[14–16] discrimination between similar but distinct contexts by active GC populations, thus, leaving the question of DGs' role in spatial segregation unresolved.

A further gap in our understanding of DGs' circuit function is the open question of whether functional lateralization exists between the left and the right DG, as observed in human and mouse CA1-3 hippocampal areas[17–20]. Particularly, do GC assemblies express hemispheric lateralization in the representation of contextual information? Correlations between BOLD signals in humans and cognitive tasks suggested that the left hippocampus supports autobiographical long-term memory formation[19], while the right one contributes to the identification of places and the egocentric navigation among them[21–23]. Data from surgeries in epileptic patients also point to lateralization, as removal of the left hippocampus affects verbal memory, and the right hippocampal resection causes deficits in complex visuospatial tasks[24,25]. Moreover, microdissection of dorsal CA1 areas in rats suggested a bias of the left hippocampus towards the emergence of memory engrams (defined as cell ensembles representing an experience (such as exploring a new environment) and which can be reactivated upon recall[26]), and of the right CA1 towards the retrieval of spatial information and route computation during navigation[27]. These studies as a whole suggest that left and right hippocampi, in both

[1]Institute for Physiology I, University of Freiburg, Medical Faculty, 79104 Freiburg, Germany. ✉e-mail: Thibault.cholvin@physiologie.uni-freiburg.de;
Marlene.bartos@physiologie.uni-freiburg.de

humans and rodents, do not carry the same information. However, data are lacking on interhemispheric functional differences in the DG. Indeed, a large body of previous studies focused on CA1-3 areas in mice, with emphasis on interhemispheric molecular, cellular and synaptic differences, for example, distinctions in receptor expression profiles in hippocampal principal cells[28], morphological and physiological single-cell characteristics including bilateral CA3 projections to CA1[29,30], and functional as well as dynamic characteristics of Schaffer collateral inputs[31–34]. In vivo studies focused on lateralization in physiological circuit function and noted differences in the properties of neuronal network oscillations[35,36] as well as in the retrieval of working memory[37] between the left and right CA3 in rats. Although functional lateralization of the DG neuronal network has so far not been observed in vivo, it is to be expected. First, the volume of the granule cell layer is larger in the right than the left DG, indicating differences in the total number of GCs providing information with their mossy fiber axons onto CA3 pyramidal cells[38]. Second, the number of GCs expressing the immediate early gene product *cfos* is higher in the left than in the right DG of mice exploring a novel environment, suggesting a higher fraction of active GCs or their elevated probability to undergo synaptic plasticity induction at their perforant path inputs in the left DG[39]. Third, it has been suggested that the left and right CA3 play different roles in the acquisition and retrieval of spatial memories, with the left CA3 being more involved in the encoding of defined locations, whereas the right one is in the integration of rout information and working memory[31]. Finally, CA3 pyramidal cells are the main target of mossy fiber synapses. This pathway is the only projection from the DG to CA3, and appears to be strictly ipsilateral[40]. Given these hemispherical differences in CA3 and the DG, it is likely that upstream GCs may be functionally lateralized.

Here, we performed two-photon calcium imaging of GC populations in head-fixed mice executing a goal-oriented task in both familiar and novel virtual environments characterized by different sets of cues and boundaries. Our results show that despite similar activity levels of GCs in both hemispheres, spatial tuning responses of GCs in the left DG convey higher information content than the ones in the right DG. Moreover, place cells in the left DG show less run-to-run variability and higher stability across days in a given environment as compared to their right counterparts. Finally, GCs with a place field remap significantly more between environments in the left than in the right DG, indicating higher contextual differentiation in the left hemisphere and a marked bias towards generalization across contexts in the right DG. Thus, we provide novel insights about the functional lateralization of memory processes taking place in the hippocampus and propose that differential routing of spatial information may underlie the emergence of hemisphere-specific context-encoding GC ensembles in the DG. This division of labor between left and right DGs would then allow the feeding of downstream hippocampal areas (e.g., CA3) with distinct contextual information associated with either segregation or generalization between distinct spatial representations[3,4,14,16].

## Results

### Imaging place- and context-modulated activity patterns of DG GCs in both hemispheres

To obtain the activity of GCs, we performed two-photon calcium imaging of GC populations with single-cell resolution. Mice ran on a spherical treadmill in one familiar and one (out of two) novel virtual environments (4 m-long virtual linear tracks; "Methods") characterized by different wall patterns, visual cues, and floor textures (Fig. 1a, e and Supplementary Fig. 1) and were offered soymilk rewards placed in distinct locations depending on the environment (Fig. 1e and Supplementary Fig. 1). To measure neuronal activity in the DG of both hemispheres, mice were injected with adeno-associated viruses (AAVs) expressing the calcium indicator GCAMP6s panneuronally in the left or right DG (Fig. 1b–d). Imaged GCs were located in the upper half of the

granule cell layer to largely record from mature GCs. We used fast scanning to record on average $234.6 \pm 25.5$ (left) and $248.6 \pm 28.5$ (right) cells per imaging session ("Methods"). After at least 10 days of familiarization to the first track, imaging sessions were started. Mice ran on the familiar track in alternation to the novel track in blocks of five runs (Fig. 1e). Animals ran slightly faster on average in the novel environment, and no difference in running speed was observed between animals with left and right-sided implantations (Fig. 1h). Licking frequency was higher inside than outside the reward zones on familiar as well as novel tracks, indicating proper identification of the reward locations in both environments (Fig. 1i and Supplementary Fig. 1).

Before analyzing activity levels and spatial information content of GCs from both hemispheres, we first computed the mean baseline noise of fluorescence signals and mean peak amplitude of calcium transients from groups of GCs jointly recorded per session. For both measures, we revealed no difference between left and right DGs (Supplementary Fig. 2). Moreover, GCs with stable place fields on subsequent runs in the familiar or novel environment were observed in both hemispheres (Fig. 1f, g). Conclusively, our analyses indicate similar somatic recording quality yielding reliable assessments of neuronal activity and spatial information content of GC ensembles in both hemispheres.

### Mean activity and spatial tuning of GCs is higher in the left than right hemisphere across days

To examine contextual representations during hippocampus-dependent learning[3,41], we imaged the same populations of GCs during 5 consecutive days in the left or right DG (Fig. 2 and Supplementary Fig. 3). The mean activity of GCs in mice exploring the familiar or the novel context was similarly sparse in the two hemispheres (Fig. 2a), and in the range of activity levels previously observed in navigating rodents[3,42,43]. Except day 1, during which GCs' mean activity in mice representing the familiar context was mildly higher than the one representing the novel context in the left DG ($P \le 0.05$; Fig. 2a), on all subsequent days, the initially observed difference was lost (Fig. 2b). This was very likely caused by a mild increase in average GC activity in the novel context on days 2–5 (Fig. 2a, right). Consequently, differences in mean activity between familiar and novel contexts were similar across days and hemispheres (Fig. 2c). The sparse representation in the DG was further reflected by small fractions of cells being active with more than 1 transient per minute in the familiar, novel or both contexts in both hemispheres (Fig. 2g). In line with the hypothesis that novel context representation emerges over time with learning[44–47], the total fraction of place cells representing the novel environment increased in both hemispheres over days (Fig. 2h, green). Notably, the fraction of all active cells was mildly but significantly higher in the left DG independent of the day of examination ($\chi^2$ test, $P < 0.001$; Fig. 2g), which was mirrored in a larger percentage of place cells (Fig. 2h), suggesting that larger ensembles of GCs are processing spatial information in the left DG. Although the number of fields represented by each place cell was similar between hemispheres, these place fields were significantly larger in the novel environment in the left DG, while no such difference was observed in the familiar one (Supplementary Fig. 4), suggesting a lower granularity in novelty space coding in the left DG. Thus, GC activity is sparse in both hemispheres with a larger place-coding fraction of cells in the left compared to the right DG.

Further analysis of spatial coding properties revealed that the average spatial information content of GCs was consistently higher in the left DG (Fig. 2d, e). Moreover, in the novel context, spatial tuning markedly increased in the left DG over consecutive days of spatial learning and reached values similar to the familiar context after the 4th day of exposure to the novel environment (Fig. 2d, right). Consequently, the difference in spatial tuning between familiar and novel contexts was particularly high on the 1st day of novelty exposure in the left DG and monotonically declined across days of learning (Fig. 2f). Thus, in line

with our previous data of improved contextual representations in the DG upon day-by-day learning[3], GCs representing the novel context showed gradually increasing spatial tuning and growing place cell populations over time. However, this effect was more pronounced in the left hemisphere, resulting in larger ensembles of GCs place cells with improved spatial encoding properties in the left than right DG.

## Place cells in the left DG discriminate better between contexts than in the right DG

Next, we probed neuronal discrimination between contexts in the left and right DG during spatial learning (Fig. 3), by first quantifying the

activity-rate difference score between contexts (activity in familiar−activity in novel) for each individual active cell based on the normalized activity-rate difference between the two contexts on each of the 5 consecutive days (Fig. 3a–c). The average activity-rate difference score was markedly higher in the left compared to the right DG across recording days ($P < 0.001$; Fig. 3c). Notably, the distribution of activity-difference scores was exceptionally broad on the 1st day of novelty exposure, indicating that although most GCs were active in only one context[14], many others were active in both contexts[3,12]. The distribution of activity-difference scores gradually narrowed on subsequent days, reaching the highest values on the last day (comparison across all days,

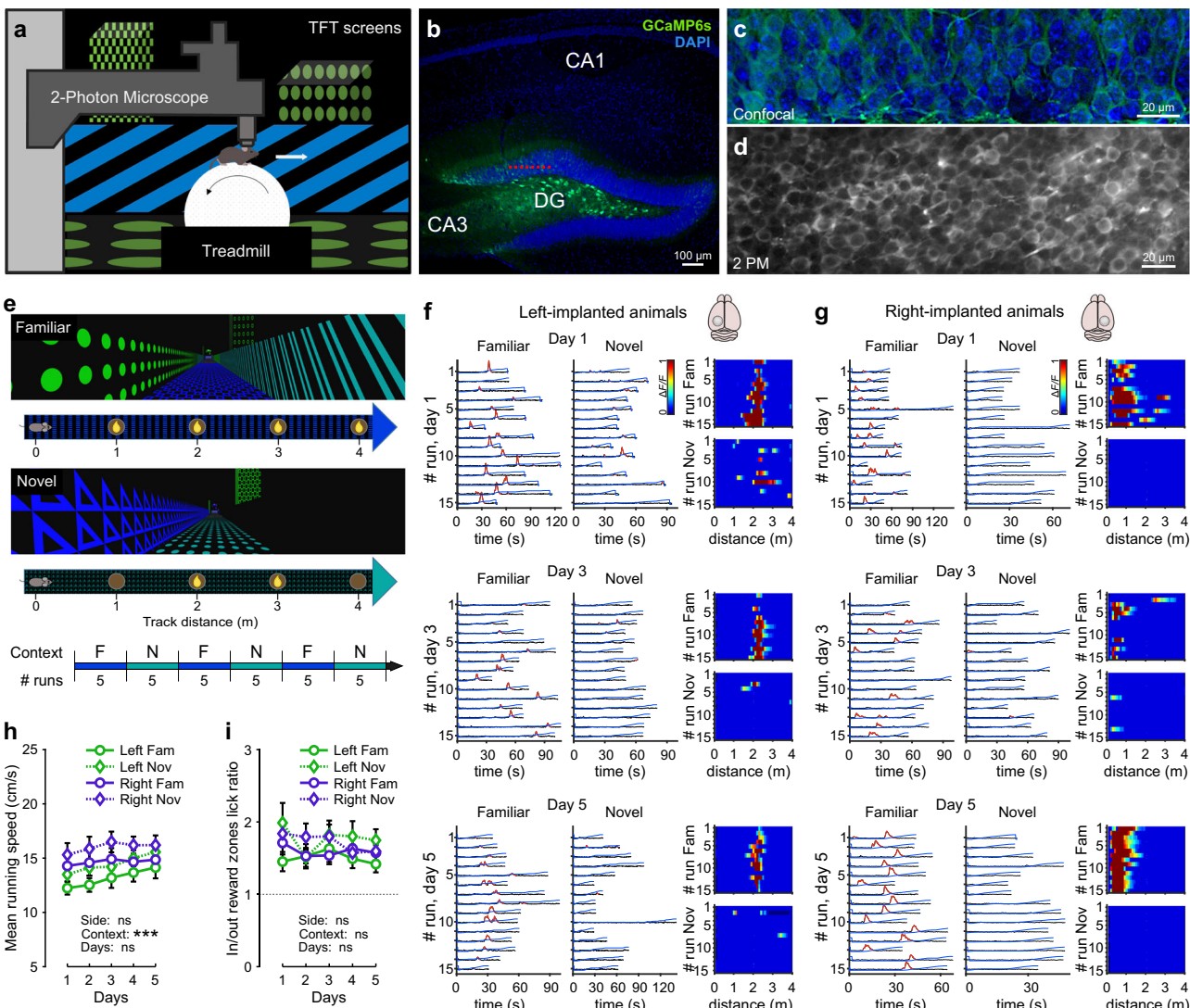

**Fig. 1 | Two-photon calcium imaging of DG granule cells activity from either left- or right-implanted mice navigating through familiar and novel virtual environments. a** Experimental schematic of our virtual reality setup for head-fixed mice. **b** Confocal image showing GCaMP6s-labeled neurons in the DG (green), tissue counterstained with DAPI (blue). Red dotted line indicates a typical imaging plane in the granule cell (GC) layer. **c** Confocal image showing GCaMP6s labeling of GCs in the DG. **d** Average fluorescence of GCaMP6s signals of GCs recorded in vivo using 2-photon microscopy imaging. **e** Top, schematic of familiar and novel 4m-long virtual environments. Yellow drops indicate the locations of soymilk rewards. Bottom, timeline of a recording session. **a**–**e** In all, 21 mice implanted either in the left ($n = 11$) or right ($n = 10$) hemisphere were used. Most animals were recorded twice, using two different novel environments, for a total of 36 datasets ($n_{left} = 18$, $n_{right} = 18$; see also Supplementary Figs. 1 and 9). **f** Left panels, raw calcium traces (gray) with significant transients (red) and linear track position (blue) over time of a

single GC from a left-implanted mouse. The same cells have been imaged over 5 consecutive days in both familiar and novel environments; days 1, 3, and 5 are shown. Right panels, calcium activity over track distance of this GC in familiar (top) and novel (bottom) environments. **g** Same as (**f**) for a GC from a left-implanted mouse. **h** Mean running speed (excluding resting periods) depending on the side of implantation and environment (familiar/novel). $n_{left} = 18$ datasets, $n_{right} = 18$ datasets. **i** Lick rates ratios observed in the reward zones (regions around the reward sites, see "Methods") compared to licking on the remaining track. A ratio above 1 indicates that mice are licking preferentially in the reward zones. $n_{left} = 11$ datasets, $n_{right} = 14$ datasets. **h**, **i** Three-way repeated measures ANOVAs, Tukey's post hoc test. ns not significant; *$P < 0.05$; **$P < 0.01$; ***$P < 0.001$. Symbols with lines indicate mean ± SEM. For exact $P$ values, see Supplementary Table 1. Source data are provided as a Source Data file.

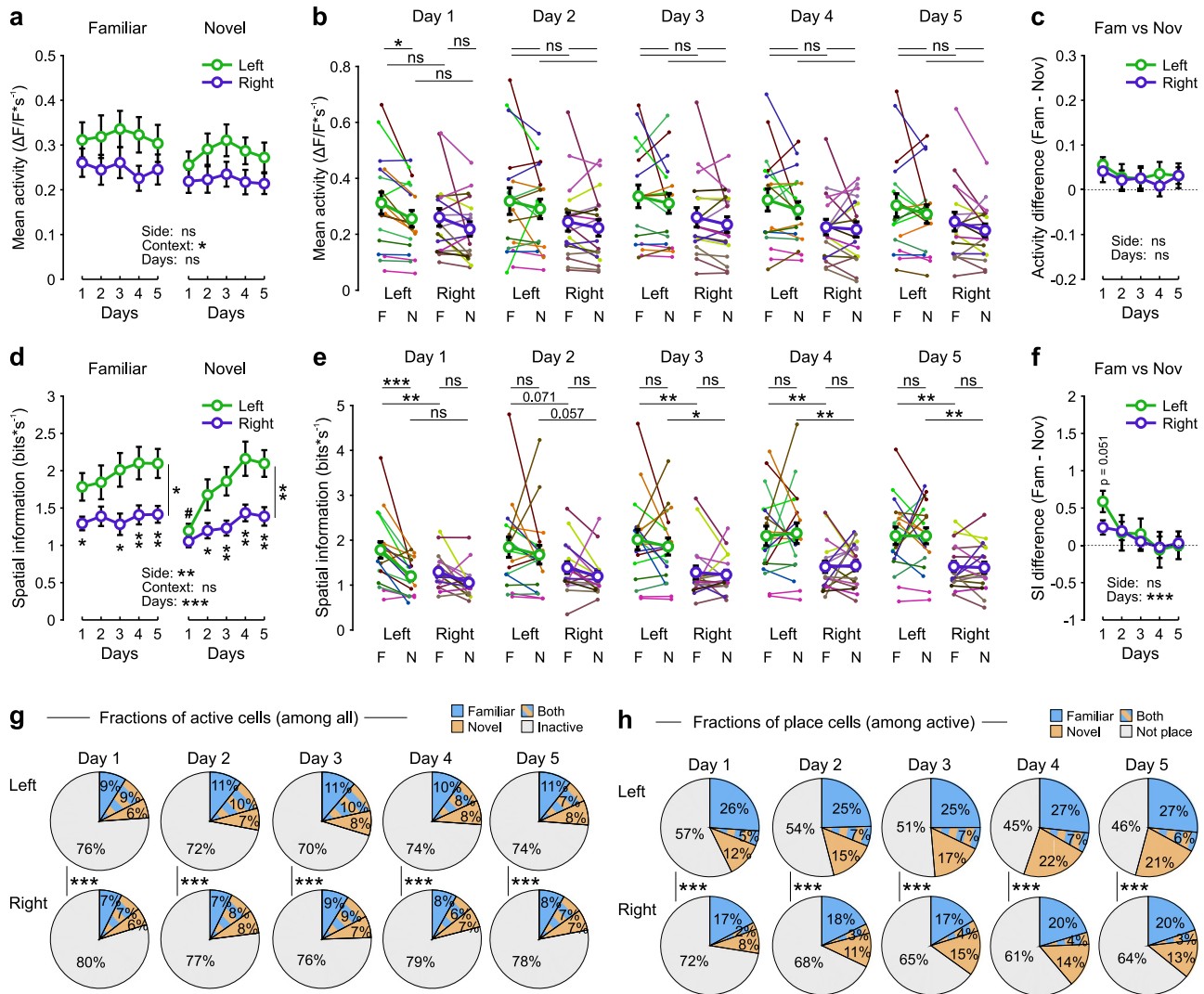

**Fig. 2 | Left- and right-hemisphere DG granule cells show differences in their activity levels and spatial information content. a** Mean activity ($\Delta F/F^*s^{-1}$) of all granule cells (GCs) from either left- and right-implanted mice recorded over 5 consecutive days in both familiar and novel environments. 21 mice implanted either in the left ($n = 11$) or right ($n = 10$) hemisphere were used. **b** Same data as (**a**) plotted as individual datasets. Each animal is represented by a different color. Thin lines connect data points (small dots) obtained in familiar and novel environments from the same animal during a given session. Thick lines connect overall mean values (circles). **c** Mean activity-difference score (activity$_{(Fam)}$ − activity$_{(Nov)}$) of all GCs. **d** Mean spatial information of active (>1 transient/min) GCs. **e** Similar to (**b**) for spatial information. **f** Mean spatial information difference score (SI$_{(Fam)}$ − SI$_{(Nov)}$) of active GCs. **g** Fraction of active cells (>1 transient/min) depending on the environment (familiar, novel or both).

**h** Fraction of place cells among the active cells depending on the environment. **a**–**f** $n_{left} = 18$ datasets, $n_{right} = 18$ datasets. **a**, **d** Three-way repeated measures ANOVAs, Tukey's post hoc test. Side, Context, and Days refer to the three main effects. **b**, **c**, **e**, **f** Two-way repeated measures ANOVAs (per day for **b** and **e**), Tukey's post hoc test. Side and Days refer to the two main effects. **g** $n_{left} = 4224$ cells, $n_{right} = 4224$ cells; **h** $n_{left} = 1009$–1255 cells, $n_{right} = 904$–1096 cells. **g**, **h** Test for population overlap ($\chi^2$ test). ns, not significant; *$P < 0.05$; **$P < 0.01$; ***$P < 0.001$. In **d**, *$P < 0.05$ or **$P < 0.01$, different from the other group (left vs right) for each individual day; #$P < 0.05$, different from the same group in the other environment on day 1 (familiar vs novel); $P$ value combined with a bar shows comparisons between left and right for all days in the familiar and the novel track. Circles with lines indicate mean ± SEM. For exact $P$ values, see Supplementary Table 1. Source data are provided as a Source Data file.

$P < 0.001$; Fig. 3c), indicating improved discrimination between contexts throughout learning. However, average discrimination tended to be higher in the left DG along the entire experiment, including the fifth recording day (left DG: median = 0.86, mean = 0.73 ± 0.012; right DG, median = 0.81, mean = 0.68 ± 0.017 on day 5; $P < 0.05$; Fig. 3c). Next, we determined trial-to-trial reliability (mean pairwise cross-correlations between all runs, see also "Methods") of place cell firing on the same track across days (Fig. 3d). Remarkably, place field reliability was markedly higher in the left DG for both tracks on every recording day (familiar, left vs right across days, $P < 0.001$; novel, left vs right DG across days, $P < 0.001$; Fig. 3d), pointing to more reliable representations in the left DG. Unexpectedly, reliability in representation increased not only for the novel context on subsequent days, as would

be expected during learning[3], but also for the familiar context (day-by-day familiar and novel, $P < 0.001$ for both comparisons; "Methods"), suggesting that novel environment exposure might influence and improve the representation of the already experienced familiar context. Finally, consistency of place field firing, defined as the activity-rate correlation between the first five runs and last five runs in a given environment ("Methods") was higher in the left DG across days (familiar, left vs right, $P < 0.001$; novel, left vs right, $P < 0.001$; Fig. 3e). Thus, GCs in the left DG show higher context discrimination and more reliable place-field representations.

To examine stability in place field locations throughout learning in the two hemispheres, we determined day-by-day activity map correlations (Fig. 3f–h). Despite the generally high

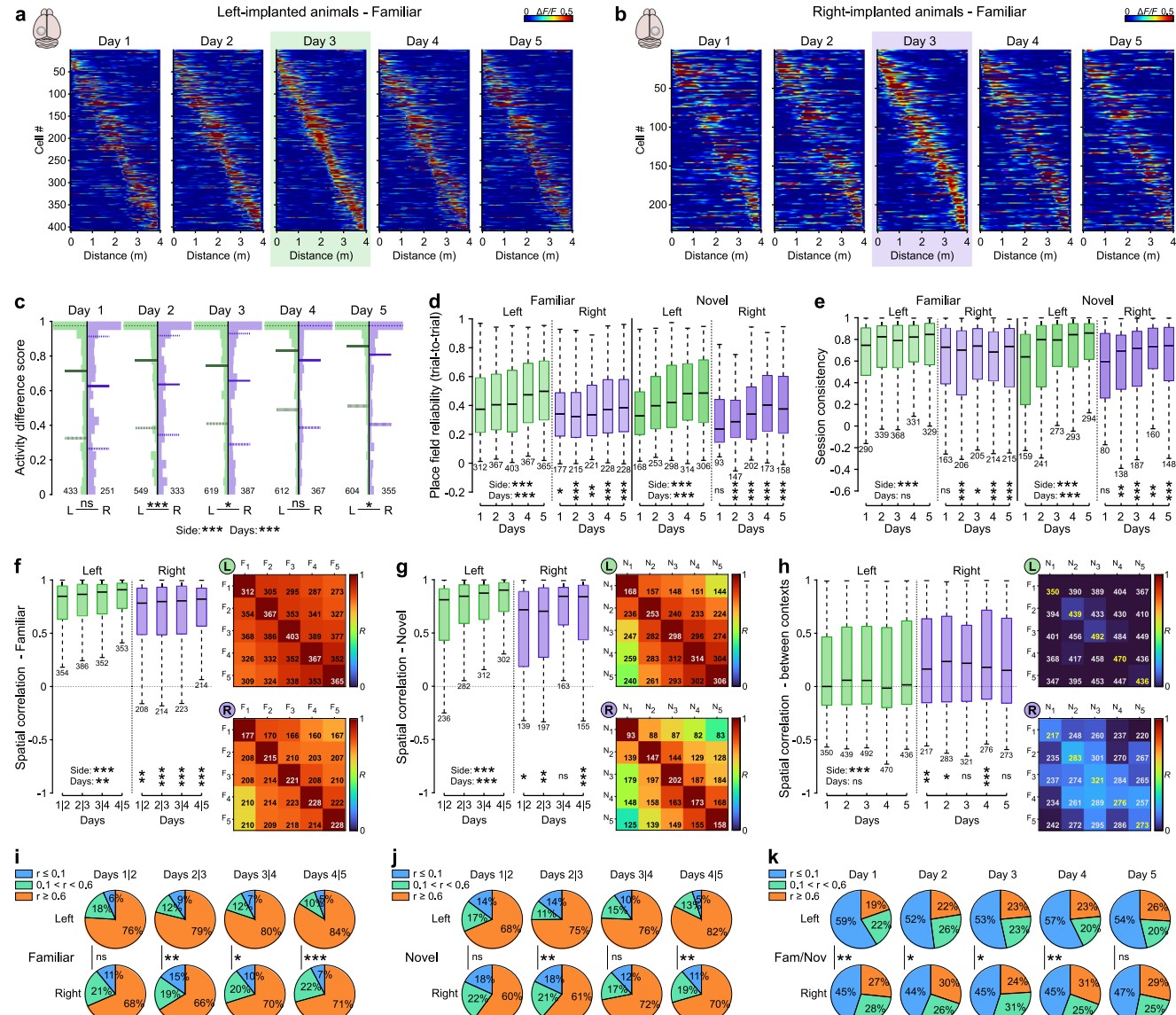

**Fig. 3 | Place granule cells of left and right DG show different levels of discrimination between familiar and novel environments. a, b** Activity maps of place GCs imaged over 5 consecutive days in left (**a**) and right (**b**) DG in the familiar environment. Sorting according to the cells activity on day 3 (21 mice in total; 11 left-implanted and 10 right-implanted mice). **c** Activity-rate difference scores ($|(\text{activity}_{(\text{Fam})} - \text{activity}_{(\text{Nov})})|/(\text{activity}_{(\text{Fam})} + \text{activity}_{(\text{Nov})})$, see also "Methods") between familiar and novel environments of place GCs. **d** Mean place field reliability (pairwise cross-correlations between all runs on the same track) of place GCs. **e** Mean session consistency (activity correlation between the first five and last five runs in a given environment) of place GCs. **f** Spatial correlation from one day to the next, familiar environment. Left panel, mean activity correlation between 2 consecutive days for place GCs, familiar environment. Right top panel, median activity map correlations (color-coded; Pearson's R) over 5 consecutive days on the familiar track (F1–F5). Each row

shows median correlation values for GCs having a place field on that day. Right bottom panel, same as the top right panel but for right GCs. **g** Spatial correlation from one day to the next, novel environment. Similar to (**f**) for the novel environment (recording days, N1–N5). **h** Spatial correlation between environments (similar to **f**, but between familiar and novel environments for each day). **i–k** Fraction of place cells (same cells as in **f**, **g**, and **h**, respectively) showing high ($r \geq 0.6$, orange), medium ($0.1 < r < 0.6$, green) or low ($r \leq 0.1$, blue) field correlations between two consecutive days in familiar (**i**) or novel (**j**) environments, and between familiar and novel environments for each day (**k**). **c–h** Two-way ANOVAs after alignment and ranking (see "Methods"), Tukey's post hoc test. **i–k** $\chi^2$ test for population overlap. Boxes, 25th to 75th percentiles; bars, median; whiskers, 99% range. Values indicate the number of cells. ns not significant; *$P < 0.05$; **$P < 0.01$; ***$P < 0.001$. For exact $P$ values, see Supplementary Table 1. Source data are provided as a Source Data file.

GC place field stability[3], activity map correlations between 2 consecutive days were significantly higher in the left compared to the right DG for both tracks (familiar, left vs right, $P < 0.001$; Fig. 3f, left; novel, left vs right, $P < 0.001$; Fig. 3g, left). Moreover, average day-by-day spatial map correlations improved across prolonged time in both hemispheres for both tracks (Fig. 3f, g, left). However, the average place field maps for both familiar and novel contexts remained more stable in the left than right DG throughout all days (Fig. 3f, g, right). Thus, stability in GC place cell maps improves

during learning to reach more stable contextual representations in the left than the right hemisphere.

Previous work led to controversial results on the amount of neuronal discrimination between environments in the DG, ranging from minimal overlap, i.e., pattern separation[4,14] to marked instantaneous overlap, i.e., generalization of spatial representations[3,13]. To address this potential controversy, we quantified place field remapping between familiar and novel contexts on consecutive days in the two hemispheres (Fig. 3h). In the left DG, activity map correlations between

contexts were close to zero throughout all days, but they were substantially higher in the right DG, pointing to improved contextual encoding in the left hemisphere (left vs right DG, two-way ANOVAs, $P < 0.001$; comparison across days, ns; Fig. 3h). This difference was even more pronounced when we focused our attention on GCs showing place fields in both contexts (Supplementary Fig. 5). Indeed, high correlations between contexts could be observed in a substantial fraction of place cells in both hemispheres, but many more place cells in the left DG had place fields in different locations (global remapping) compared to the right DG (Supplementary Fig. 5h, left; see also Supplementary Fig. 5a, b for place maps on familiar and novel track). We thus quantified the fraction of globally remapping and stable (i.e., generalizing) place cells on every recording day (Supplementary Fig. 5k). The fraction of globally remapping place cells (mean place field correlation between tracks $r \leq 0.1$) was significantly higher in the left DG, whereas the fraction of place cells with stable place fields across contexts (mean place field correlation between tracks $r \geq 0.6$) was markedly higher in the right DG (Fig. 3i–k and Supplementary Fig. 6). Finally, we verified that considering only the first dataset from each animal (initial novelty exposure, *novel 1*) led to similar results (Supplementary Fig. 7). Thus, consistent with the higher space and context specificity of the left DG, a larger space-coding GC ensemble appears to be specific to each context in the left hemisphere. In contrast, and in line with the observed lower space and context specificity of the right DG, a larger GC population seems to generalize between contexts in the right hemisphere.

### Context representations are more reliable in the left than right DG

To address whether the observed differences in activity rates and spatial correlation-dependent parameters between left and right DG may influence the accuracy of encoding of space and context, we applied an inverse approach and used the recorded cell activity to decode the animals' concurrent location and context with a population-vector-based method (refs. 42, 48; Fig. 4 and Supplementary Fig. 8; "Methods"). We used subsamples of randomly selected active cells of varying size (range: 5–100 cells) recorded in a given session, to first define decoding performance for both context and space independent of the hemisphere (Fig. 4a). With increasing number of cells, the context and spatial error monotonically declined (Fig. 4b–e). Consistent with our experimentally observed higher ability to differentiate between contexts in the left DG, context and spatial errors were higher for the right than the left hemisphere for all recording days (Fig. 4b–e). Notably, average context and spatial errors for a fixed number of cells ($n = 50$) declined across days and reached the lowest values in the left DG (Fig. 4c, e). A similar trend could be observed if we restricted our analyses to sessions in the familiar or novel context (Supplementary Fig. 8d–k). Here, interestingly, the average context (Supplementary Fig. 8j, k) and spatial (Supplementary Fig. 8f, g) decoding errors predicting the novel environment were similar between left and right DG on day 1, but differences emerged over the course of subsequent days of novelty exposure, supporting the hypothesis that learning improves spatial and contextual representation, particularly in the left DG. Thus, the left hemisphere is carrying more information about space and context than the right one.

To systematically examine which of these activity-dependent spatial and contextual coding parameters may account for the observed differences in decoding quality between left and right DG, we sorted GCs in each dataset by their value of a given parameter and separated half of the cells with the higher values (upper half) from the remaining cells (lower half). We then divided the decoding error obtained from the upper half by the one obtained from the lower half for each dataset (Supplementary Fig. 8l–o). A ratio markedly below one indicates that the examined parameter had an influence on decoding performance. This approach revealed that the mean activity rate and

spatial information (SI) were strong determining factors of spatial and contextual decoding performance (Supplementary Fig. 8l, n). Notably, the predicting values for these spatial and contextual coding properties were always higher in the left than right DG, and while they improved over days in both hemispheres, the highest improvement was observed in the left DG (Supplementary Fig. 8m, o).

The hippocampus is required for rapid context recognition during behavior[49,50]. The readout of this information depends, however, on the duration of contextual exposure. We, therefore, investigated how rapidly reliable contextual information can be read out from hippocampal GC ensembles in the left and right DG of varying sizes (Fig. 4f, g). We quantified the decision value for the identification of a context for a given neuronal ensemble size over a continuously increasing integration time (Fig. 4f). As expected, the decision accuracy improved with increasing population size and prolonged time of observation. However, the time interval needed to reach 90% of performance accuracy for a similarly sized GC population (50 cells) was significantly higher in the right compared to the left DG (Fig. 4g). Moreover, the integration time declined over days, indicating that day-by-day exposure to the virtual environments improved contextual representation. Indeed, the median time required to achieve 90% contextual decoding accuracy was 30 s in the left and 65 s in the right DG on day 1, and improved to 9 s in the left and 30 s in the right DG on day 5 (Fig. 4g). Thus, a relatively small GC ensemble from the left DG can provide a reliable representation of the currently explored environment much faster than a similar cell population from the right hemisphere.

## Discussion

Using 2-photon calcium imaging, we obtained GC population activity in the left and right DG and found that activity patterns representing the familiar and the novel environments differed markedly more in the left than in the right hemisphere, resulting in higher context specificity in the left DG (Fig. 3c, h). Moreover, trial-to-trial reliability and session consistency were higher in the left than right DG (Fig. 3d, e), thereby permitting more stable spatial and contextual representations, which support the formation and recall of episodic memories[2,51,52]. Finally, cell associations representing novel contexts increased across learning and resulted in larger GC place cell ensembles in the left DG, suggesting bilateral differences in the sparsity of representations (Fig. 2g, h). Thus, we provide first evidence for hemisphere-specific spatial and contextual representations in the DG ranging from stronger discrimination (pattern separation) in the left, to weaker discrimination between environments (generalization) in the right DG (Fig. 4h). These data may help to reconcile current inconsistencies in the literature reporting different degrees of spatial discrimination on the level of the DG network[3,4,6,13,14,42].

Functional lateralization is common in vertebrate brains and proposed to provide benefits in sensory, cognitive, and motor capabilities, presumably by allowing parallel and separate processing of information by the two hemispheres[31,53–55]. For example, left-right cortical asymmetry seems to improve the ability to search for food while simultaneously being attentive to predators[56–59]. Hippocampal functional lateralization is well-documented in humans[19,20], but has been less investigated in rodents. Previous work largely focused on the lateralization of the CA1 and CA3 neuronal network and showed differences in anatomical connectivity, spine size and the expression of synaptic plasticity[28,31,33,60]. CA1 pyramidal cells receive extensive CA3 afferents from both the ipsi- and contralateral sides[29,30]. However, inputs from the left CA3 form smaller synapses and show stronger associative long-term plasticity than inputs from the right CA3[28,32–34,60]. Consistent with the finding that optogenetic silencing of the left but not the right CA3 impaired the performance in a long-term spatial memory task[34], it appears that the two hippocampal hemispheres do not provide the same information to CA1. Indeed, it has been

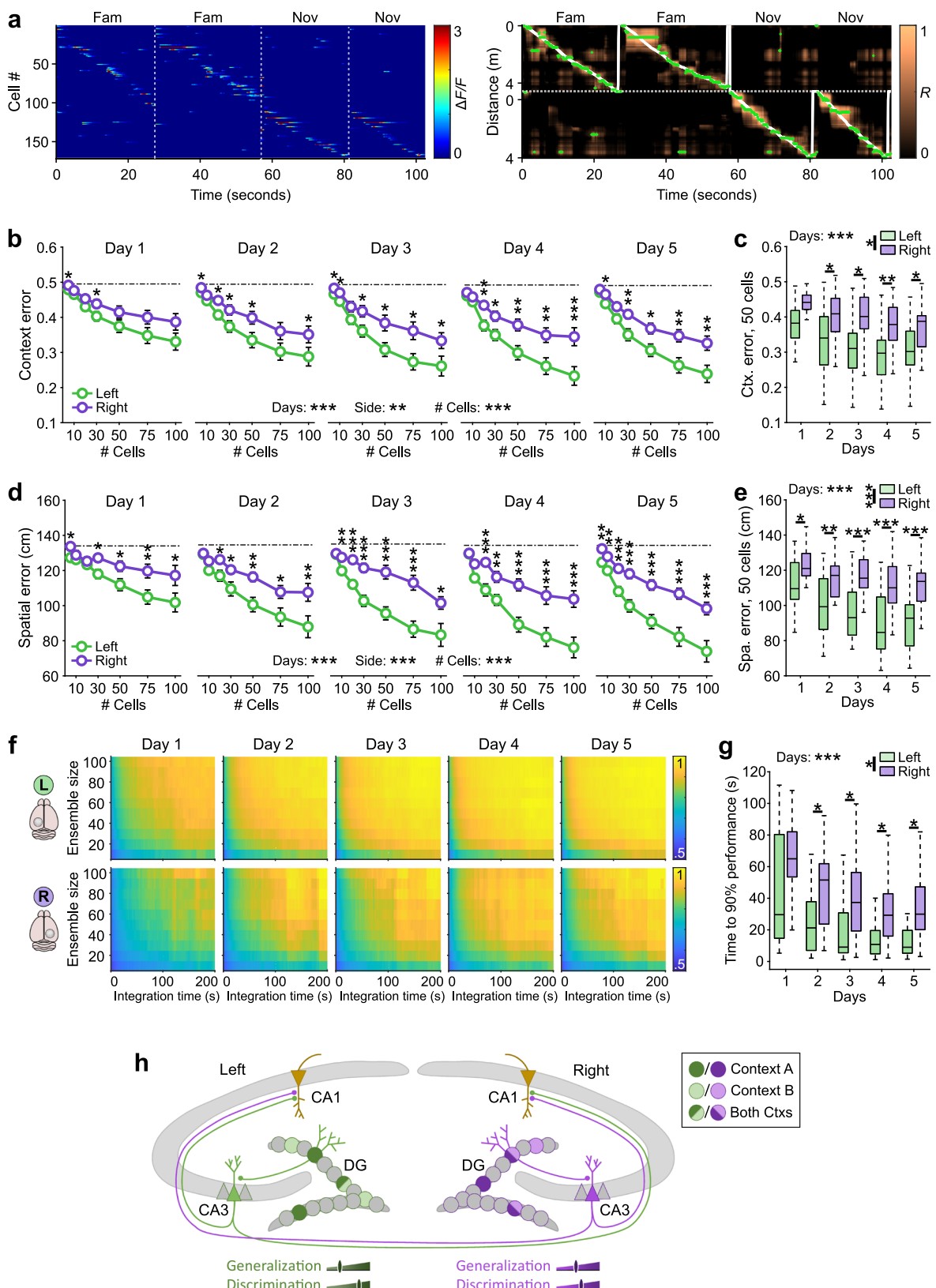

suggested that the left CA3 is more responsible for the storage and discrimination of discrete locations within the environment, whereas the right CA3 seems to be more involved in integrating contextual information, presumably to guide navigation during spatial working memory tasks[31,34,61,62]. This information is transmitted to CA1, where bilateral data are merged[17] before being relayed to the cortex to

support behavioral execution[63–65]. Our work fits to the proposed functional lateralization of CA3 by demonstrating that interhemispheric differences in the formation and retrieval of spatial and contextual memory exist already upstream of CA3, in the DG. As direct bilateral connections from GCs to CA3 pyramidal neurons are lacking[66], strict unilateral mossy fiber projections may support

**Fig. 4 | Decoding of space and context from neuronal activity confirms interhemispheric differences in the encoding of familiar and novel environments.** **a** Decoding example. Left panel, activity over time of an ensemble of GCs recorded in the left hemisphere. Right panel, decoder output. White line denotes true position of the mouse and the green dots the most likely decoded position. **b** Contextual decoding error as a function of the number of cells used simultaneously for decoding. **c** Average contextual decoding errors for ensembles of 50 cells. **d** Same as (**b**) for spatial decoding error. **e** Average spatial decoding errors for ensembles of 50 cells. **f** Cumulative probability for decoding the correct environment as a function of time and ensemble size for GCs recorded in the left (top) and right (bottom) hemispheres. Bright colors indicate high predictive value. Note, that high predictive values can be obtained with smaller ensembles in less time in the left than right DG. **g** Average time to 90% context-decoding accuracy for a fixed ensemble size of 50 cells. **h** Schematic: left GCs show low generalization and high discrimination between contexts, while right GCs offer less context selectivity. **b**–**g** $n_{left}$ = 18 datasets, $n_{right}$ = 18 datasets. **b**, **d** Three-way repeated measures ANOVAs, Tukey's post hoc test. **c**, **e**, **g** Two-way repeated measures ANOVAs (for each parameter independently in **c** and **e**), Tukey's post hoc test. Boxes, 25th to 75th percentiles; bars, median; whiskers, 99% range. ns, not significant; *$P$ < 0.05; **$P$ < 0.01; ***$P$ < 0.001. Stars above individual circles refer to comparisons between cells in the left and right DG (**b**, **d**). # cell *** refers to a three-way ANOVA (influence of cell number, side, and day). Circles with lines indicate mean ± SEM. For exact $P$ values, see Supplementary Table 1. Source data are provided as a Source Data file.

functional CA3 lateralization. However, future work will be required to shed light on potential differences in morphological and physiological characteristics of mossy fiber synapses originating from the left and right DG, including bouton size, extent of synaptic plasticity, receptor subunit composition, and connectivity between GCs and CA3 pyramidal cells.

What are the potential mechanisms that could account for the observed differences in spatial and contextual representation across hemispheres in the DG? The entorhinal cortex (EC) provides both self-motion- and global-feature-based environmental information to the hippocampus[67–70]. Optogenetic modulation of excitatory projections from the EC to the hippocampus influence the encoding of spatial and environmental features within CA1 and CA3 pyramidal cells[71–73]. Thus, interhemispheric differences in the projections from the EC to the hippocampus may underlie the observed lateralization of spatial representation and context discrimination of the left and right DGs. Moreover, differences in the divergence and the strength of lateral inhibition provided by local GABAergic interneurons to the DG circuitry[74–76] may contribute to hemisphere-specific shaping of GC place field characteristics. Finally, considering the marked anatomical and functional lateralization of CA3 principal cells (see above), it appears likely that back-projecting CA3 principal cells[77,78] transmit different information to the left and right DGs. Thus, future work will have to address the contribution of excitatory and inhibitory inputs to the DG on the functional lateralization observed at the level of the granule cells.

What might be the functional relevance of stronger decorrelation in the left and weaker discrimination of environmental representations in the right DG for downstream hippocampal areas? During environmental exploration, a spatial map progressively emerges within the DG[79], which pushes downstream CA regions either to recall a familiar or to form a novel representation[4]. Whether a familiar (already stored) representation is updated or a novel one is formed may depend on the influence of both DG inputs on the attractor dynamics of downstream hippocampal areas. Additional inputs, such as the modulatory signals originating from parahippocampal structures providing information about the saliency of novel environments[80] and the ones from the EC may further contribute to the determination whether CA1 networks will lean towards the refinement of an already stored memory or the creation of a new representation. Modeling studies propose that nonlinear interactions between inputs from the DG, neuromodulators and the EC are required for memory recall[81]. Thus, we propose that inputs of the two DGs may influence the switch between the recall of an existing memory and the formation of a novel representation. Evidently, environmental parameters such as different settings of virtual realities and different degree of their alterations may also be determining factors in this process.

Our work reconciles the prevailing debate on the different roles of the DG in orthogonalizing cortical inputs to support cognitive discrimination of distinct representations, and in generalizing shared properties of distinct memories[4,14,15,82,83]. We show that a substantial fraction of GCs represents context-specific representations, particularly in the left DG, which in turn will support contextual discrimination processes in the hippocampus[6]. Our work also confirms earlier evidence that a substantial fraction of GCs might encode common features of different spatial contexts and thereby corroborates context generalization[3]. We show that this fraction of cells is particularly located in the right DG (Fig. 3 and Supplementary Fig. 5). Moreover, transient increase in excitability of GCs ensembles expressing the immediate early gene *cfos* during initial learning improves context discrimination[84,85], presumably by increasing *cfos*-driven functional and structural plasticity. Consistent with this finding, *cfos*-positive GC populations are larger in the left compared to the right DG[39]. Thus, we propose that a subpopulation of GCs particularly numerous in the left DG may be crucial for context discrimination, whereas a separate ensemble of context-invariant GCs predominantly located in the right DG may be involved in context generalization[3,12,13]. Taken together, our results suggest an intriguing connection between DGs' functional lateralization and the published inconsistencies in the degree of contextual discrimination in the DG of behaving mice[3,4,13,14,42]. More work is needed to rigorously test how the observed interhemispheric DG differences may be influenced by the richness of the virtual realities as well as the behavioral relevance of the task[14], and to bridge the gap between functional DG lateralization and the physiological properties of the involved GCs.

## Methods

### Subjects

All experiments involving animals were carried out according to national and institutional guidelines and approved by the "Tieversuchskommission" of the Regierungspräsidium Freiburg (license #G20/137) in accordance with national legislation. We used a total of 21 C57BL/6J wild-type male mice aged 9–12 weeks at the beginning of the experiments. 11 mice were implanted in the left hemisphere, 10 in the right one. Most animals were recorded twice, using two different novel environments to obtain two independent datasets per animal (for a total of 36 datasets, 18 left, 18 right), allowing us to reduce the number of animals used. Mice were housed on a 12-h light–dark cycle in groups of 2–3 mice in a room maintained at a temperature of 21 °C (±1 °C) and relative humidity of 55% (±10%). No statistical methods were used to predetermine the sample size. The experiments were not randomized and the investigators were not blind to allocation during experiments and outcome assessment.

### Surgery: virus injections and head plate implantation

All surgical procedures were performed in a stereotactic apparatus (Kopf instruments) under anesthesia with 1.5–2% isoflurane and analgesia using 0.1 mg kg$^{-1}$ buprenorphine. An eye lubricant ointment (Bepanthen, Bayer) was applied to protect corneal membranes during surgeries. The skin was first disinfected twice with alternating solutions of 70% ethanol and Povidone-iodine (Betadine, Avrio Health L.P.) before the surgical incision. A small craniotomy (diameter 0.5–1 mm) was made over the hippocampus (A/P −2.0 mm, M/L ±1.4 mm from Bregma). Dura was carefully pierced with a needle in the center of the

craniotomy and a glass micropipette was lowered at D/V −1.75 mm below the brain surface, where 400 nL of AAV1.Syn.GCaMP6s.W-PRE.SV40 (titer $1 \times 10^{12}$ vg (viral genomes) per ml; Addgene plasmid #100843[86]) were slowly injected in the dentate gyrus. The target volume was slowly injected over ~2 min and the glass micropipette was further left in situ for 7 min to ensure complete diffusion of the viral vector in the parenchyma. During the same surgical procedure, mice were implanted with a stainless-steel headplate ($25 \times 10 \times 0.8$ mm with 8 mm-wide central aperture) installed horizontally over the hippocampus, centered on A/P = −2.0 mm and M/L = ±1.8 mm from Bregma and secured with dental cement (Super-Bond Universal Polymer Radiopaque, Catalyst V and Quick Monomer, Sun Medical). Postoperative analgesic treatment consisted of Carprofen administration (5 mg kg⁻¹ of body weight) provided during 3 days after surgery. Mice were allowed to recover from surgery for at least 5 days before any further experiment.

### Surgery: imaging window implantation

During a second surgery taking place at least 7 days after the first one (described in the previous section), an imaging window was implanted. Using the same anesthesia/analgesia protocol as described above, a craniotomy (diameter 3 mm) was drilled at A/P −1.9 mm, M/L ±1.8 mm. Under continuous irrigation with chilled saline, part of the somatosensory cortex and posterior parietal association cortex located above the hippocampus were progressively aspirated until the external capsule was exposed. The outer part of the external capsule was then gently peeled away using fine forceps, leaving the inner capsule and the hippocampus optically accessible, yet undamaged. The imaging window implant consisted of a 3-mm-wide coverslip (CS-3R, Warner Instruments) glued to the bottom of a stainless-steel cannula (3-mm diameter, 1.3-mm height). This window was gently lowered into the craniotomy using forceps until the coverslip was sitting on the external capsule. The implant was then fixed to the surrounding skull using cyanoacrylate. Mice were allowed to recover from window implantation for at least 4 days before any further experiment.

### Virtual environment setup

As previously described[3,42,87,88], our custom virtual environment setup consisted of an air-supported polystyrene ball (20-cm diameter) attached on one side with a small metal axle (restraining the ball motion to the forward–backward direction). Ball movement was monitored using an optical sensor (G-500, Logitech) and translated into forward motion inside the virtual environment. The forward gain was set such that 4 m of distance traveled along the circumference of the ball equaled one full traversal of the linear track. When the mouse reached the end of the track, screens were blanked for 4–10 s before the mouse was "teleported" back to the start of the linear track. The virtual environment was displayed on four TFT monitors (19″ screen diagonal, Dell) arranged in a hexagonal arc around the mouse and placed ~25 cm away from the head of the animal, thereby covering ~260° of the horizontal and ~60° of the vertical visual field of the mouse. The virtual environments were created and simulated using the open-source 3D rendering software Blender 2.79b (available at www.blender.org). The three different environments used in the present work consisted of distinct arrangements of textured walls, floors and 3D-rendered objects placed along the tracks sides. When the mouse reached any of the rewarded sites (of which the positions were different for each environment; see Fig. 1e and Supplementary Fig. 1), 2 μl of soymilk were dispensed through a spout in front of the mouse.

### Behavioral training

Five to seven days after head plate implantation, mice were allowed to explore a first (hereafter-called "familiar") virtual environment for 10–30 min daily, with gradually increasing timespans over days. Once the mice showed evidence of habituation to this behavioral task (i.e.,

appropriate position on the ball and consistent voluntary running, usually after 5–10 days of training), food scheduling was initiated with a goal of ~85% of the ad libitum body weight. Training in the familiar environment was maintained for 30–60 min daily until consistent reward licking and voluntary running were observed (i.e., at least 10–12 days of exposure to the familiar environment before any imaging session). For half of the mice, this familiar environment was the one indicated as 'familiar' in Fig. 1e. For the other half of the animals, the actual familiar environment was the one presented as "novel" in Fig. 1e (and thus, for these animals, one of the two novel environments was the one presented as "familiar" in Fig. 1e). In both configurations, and therefore for all mice, the familiar environment was always offering four reward sites, while the two novel environments contained two reward sites each (in distinct locations, see Fig. 1e and Supplementary Fig. 1).

### Behavioral paradigm for imaging sessions

From the first day of imaging, mice were introduced to a novel environment, which had different visual cues and floor and wall textures, but had the same dimensions as the familiar environment. For each imaging session, mice alternatively ran on the familiar and one of the two novel tracks for a total of 30 runs per day. Runs were grouped by blocks of 5x each track (starting with the familiar one) for a total of 3 blocks and 15 recordings for each track. This recording procedure was repeated over 5 consecutive days during which the same exact field of view (and therefore the same population of cells) were imaged. In many of the mice, the visible area under the imaging window was sufficiently large to allow for the selection of several imaging fields of view that contained different populations of granule cells. We, therefore, replicated this procedure a second time for most of the animals using a second novel environment. The familiar environment always remained the same for a given animal. In this manner, we performed, in total, 36 experiments of 5 days each (18 experiments in animals implanted on the left hemisphere and 18 in animals implanted in the right hemisphere). Among the 18 datasets obtained from left-implanted animals, 11 datasets were acquired using the first novel environment, and 7 using the second novel environment. Similarly, among the 18 datasets obtained in animals implanted on the right hemisphere, 10 were acquired using the first novel environment, and 8 using the second one. Grand totals of granule cells imaged were of 4224 granule cells in left-implanted animals and 4475 granule cells in right-implanted animals (Mean number of cells imaged per session ± SEM: left, 234.6 ± 25.5; right, 248.6 ± 28.5).

### Reward-related licking behavior

Licking by the mice was monitored using an infrared optical lick detector placed in front of the metal lick spout dispensing the reward. For some of the recording sessions, no lick data were acquired. In total, lick data from 11 out of 18 sessions in the left-implanted animals and 14 out of 18 sessions in the right-implanted animals were recorded. The reward zones were defined as the 5 bins (i.e., 25 cm) around the center of the reward sites. The remaining of the track was considered as outside of the reward zones. Thus, the ratio of licking between the inside and outside of the reward zones was computed as the mean lick rate in the reward zones divided by the mean lick rate on the remaining track.

### In vivo two-photon calcium imaging

In vivo calcium imaging was performed using a resonant/galvo high-speed laser scanning two-photon microscope (Neurolabware) through a ×16 objective (Nikon, 0.8 N.A., 3 mm WD) with a frame rate of 15.5 Hz and using a single plane for imaging. GCaMP6s was excited at 930 nm with a femtosecond-pulsed two-photon laser (Mai Tai DeepSee, Spectra-Physics). To block ambient light from reaching the photodetectors, the animal's head plate was attached to the bottom of an

opaque imaging chamber before each experiment, and the mouse was then affixed to the virtual environment setup using this head chamber. A ring of black foam rubber was placed between the imaging chamber and the microscope objective, and a metallic collar surrounding the objective was sitting on the imaging chamber, blocking any remaining stray light. Granule cells were imaged at a depth ~650 μm in the DG (Supplementary Fig. 9). Laser power and photomultiplier (PMT) detectors (Hamamatsu H11706-40 GaAsP) were compensated appropriately for each imaging session, ensuring consistent recording conditions. Data were acquired using the Scanbox software (Neurolabware).

## Histology

At the end of the experiments, mice were deeply anaesthetized using a mixture of ketamine/xylazine (Sigma Aldrich), then intracardially perfused with 0.1 M phosphate-buffered saline (PBS) for 5 min followed by 4% paraformaldehyde (PFA) in PBS for 10 min. Brains were further immersed in 4% PFA for 3 h, then kept in PBS until they were cut into 80-μm-tick coronal sections containing the hippocampus (usually the day after). Slices were counterstained with DAPI and mounted in Mowiol. Image stacks of GCaMP6s and DAPI fluorescence were acquired with a confocal microscope (LSM 710, Zeiss). In all animals, the locations of the in vivo-imaged hippocampal regions were confirmed by comparing averaged two-photon calcium images with confocal images.

## Calcium imaging data processing and ROI extraction

The processing of all raw calcium data was done using the Python-based toolbox Suite2p, a free automated pipeline for processing two-photon calcium imaging recordings (available at www.github.com/Mouseland/suite2p). Briefly, Suite2p first aligns all frames of a calcium movie using two-dimensional rigid registration based on regularized phase correlation, subpixel interpolation, and kriging[89,90]. This toolbox then allows visual inspection of the registered movie. Only datasets in which consistent alignment over the entire course of the experiment (i.e., 5 days of recording) was achieved were kept for further processing. Suite2p then performs automated cells detection and neuropil correction by computing a low-dimensional decomposition of the data, which is used to run a clustering algorithm that finds regions of interest (ROIs) based on the correlation of the pixels inside them. All ROIs were manually curated to ensure the most accurate selection of granule cells and care was taken to verify that segmented entities were clearly visible throughout the entire experiment. Granule cell somata were mainly identified based on size and location: granule cells can be disambiguated from other cell types by their small soma size and their location in the granule cell layer. Neurons with unusually large somata or locations within the hilus were discarded, and the frequency and shape of the calcium transients were also used to discard any putative interneuron.

Significant calcium transients were identified as previously described[87,91]. This approach has been used in numerous hippocampal in vivo calcium imaging studies in other[88,92,93] and our lab[3,42]. Imaging was performed in the upper blade of the granule cell layer (corresponding to coordinates AP = 2.0 mm, ML = 1.2 mm, DV = 1.8 mm from brain surface and 600–700 μm under the window) in both left- and right-implanted animals, thereby ensuring equivalent imaging locations along the lateral and septo-temporal axes of the DG. Moreover, the imaging plane of the field of view always comprised GCs situated superficially, in the middle and in the deep granule cell layer, thus our data included cells imaged along the entire radial axis of the granule cell layer. We restricted our analyses to periods with a running speed of at least 5 cm s⁻¹. In brief, calcium traces were corrected for slow changes in fluorescence by subtracting the 8th percentile value of the fluorescence-value distribution in a window of 20 s around each time point from the raw fluorescence trace. We obtained an initial estimate

on baseline fluorescence by calculating the mean and standard deviation (s.d.) of all points of the fluorescence signal that did not exceed 2.3 s.d. of the total signal. We then divided the raw fluorescence trace by this value to obtain the ΔF/F trace. This trace was used to determine the parameters for transient detection that yielded a false positive rate (defined as the ratio of negative to positive oriented transients) <5% and extracted all significant transients from the raw ΔF/F trace[87]. Definitive values for baseline fluorescence and baseline s.d. were calculated from all points of the trace that did not contain significant transients. A transient mask was created, and for further analysis, all values of this ΔF/F trace that did not contain significant calcium transients were set to zero[87] in order to improve the signal-to-noise ratio. Using this method, ΔF/F is expressed in units of s.d. (the standard deviation of the baseline fluorescence).

## Activity differences and spatial information

Activity-rate difference scores were calculated either for each dataset (and all cells) as the difference between the activity in the familiar and the novel environment (activity$_{(Fam)}$ – activity$_{(Nov)}$; Fig. 2c, f), or for each place cell using the following formula: $|$(activity$_{(Fam)}$ – activity$_{(Nov)}$)$|$/(activity$_{(Fam)}$ + activity$_{(Nov)}$), in Fig. 3c and Supplementary Fig. 5c. To measure the spatial information (SI) content, we adapted a common method of SI assessment[94] for calcium imaging data. The average calcium activity (mean ΔF/F) was computed for each 5-cm-wide bin along the linear track and used as an approximation for the neurons' average firing rate in that location. As previously described[3,42], spatial information was calculated as $SI = \sum_{i=1}^{N} \lambda_i \log_2 \frac{\lambda_i}{\lambda} p_i$ in which $\lambda_i$ and $p_i$ are the average calcium activity and fraction of time spent in the i$^{th}$ bin, respectively, $\lambda$ is the overall calcium activity averaged over the entire track, and $N$ is the number of bins on the track (80 bins in total). Therefore, SI content is inferred from differences in the calcium activity and expressed as bits * s⁻¹. For each cell, significant SI was assessed by shuffling $y$ traces (position of the animal along the track) of the original dataset and computing the SI score of the resulting shuffled dataset. This procedure was repeated 1000 times, and the $P$ value was determined as the fraction of shuffled datasets in which the SI score was higher than the SI score of the original dataset. Spatial information was considered significant if $P < 0.05$.

## Place-field identification

Place fields were identified according to published methods[3,42,87,88]. In brief, the mean ΔF/F was computed from significant calcium transients for each 5-cm-wide bin along the linear track (80 bins) and this mean fluorescence over distance was then smoothed by averaging over the three adjacent points for each bin. Potential place fields were initially identified as contiguous regions of this ΔF/F over distance plot in which all of the points were greater than 25% of the difference between the bin with the highest ΔF/F value and the baseline value (mean of the lowest 20 out of 80 bins' ΔF/F values). In addition, the candidate place fields had to fulfill the following criteria: (1) the width of the potential field had to be of at least 3 bins (corresponding to 15 cm running distance); (2) the mean ΔF/F value inside the field had to be at least seven times the mean of the ΔF/F value outside the field; and (3) significant calcium transients had to be present at least 20% of the time in which the mouse was moving in the field. Potential place fields that fulfilled these criteria were accepted if their $P$ value from bootstrapping exceeded 0.05. For bootstrapping, the ΔF/F trace for each experiment was broken into segments of 50 consecutive imaging frames and randomly shuffled, and this was performed 1000 times. Then the place field detection procedure described above was performed on each of the shuffled ΔF/F traces, and the $P$ value of the place field was defined as the number of these randomly shuffled traces on which a place field was detected according to the outlined criteria divided by the number of shuffles (i.e., 1000).

### Place field consistency, the similarity between contexts and trial-to-trial reliability

To assess the similarity of a place cell spatial representation in different environments, we calculated the mean ΔF/F value for each of the 80 bins on the track, based on all significant calcium transients (activity map) for each cell and environment. As previously described[42], each recording session consisted of three blocks of five runs in each environment (for a total of 6 blocks and 30 runs). Therefore, the stability of place fields was measured as the cross-correlation of the mean activity maps between the average activity of the first and the second (Fig. 4f, h) or third (Fig. 3e and Supplementary Fig. 5e) blocks of five consecutive runs on the same track and session. The similarity of place fields between contexts (remapping) was quantified as the correlation of mean activity maps for all runs in familiar and novel environments (Figs. 3h and 4f, h and Supplementary Fig. 5h). Finally, the trial-to-trial reliability was computed by calculating the pairwise cross-correlations between the calcium signals of all individual runs in one session on the same track and then averaging the obtained values for each cell (Fig. 3d and Supplementary Fig. 5d).

### Population-vector-based decoding

As described previously[42], to decode position and context from neuronal activity data, we first split every dataset in two interleaved halves of template- and testing runs, respectively. Templates for population-vector-based decoding were then generated using the template runs by calculating the mean activity for each 5 cm bin on familiar and novel linear tracks. Neuronal activity from the testing data was used to calculate population activity vectors for each 100-ms bin, and we computed the Pearson correlation value for each of those population vectors with the template population vectors for each position. The most likely decoded location was then determined as the spatial bin that had the highest correlation with the population activity at a given time[48]. Context error for each time bin was either 0 if the decoded location was in the correct environment, or 1 otherwise (Fig. 4a, right). The mean context error was then obtained by averaging over all time bins (Fig. 4b, c). To obtain the mean spatial error, we calculated the absolute distance between the most likely decoded spatial location (irrespective of the decoded context) and the true location at each time point and averaged this distance across all time points (Fig. 4d, e). The same approach was used to determine context and spatial errors in each environment (familiar or novel, Supplementary Fig. 8d–k).

### Cumulative context-decoding performance

Contextual decoding was often incorrect in individual 100-ms time bins, although on average it correctly predicted in which environment the animal actually was in all recordings (Fig. 4b, c). We hence reasoned that averaging over an increasingly larger number of 100 ms bins should increase the likelihood of decoding the correct context. To get a robust estimate for the time course over which such improvement might happen, we randomly drew 50 cells from each recording to perform context decoding as outlined above. We then calculated a decision value for the current context for increasingly longer time intervals (Δt). For each Δt, we divided the test data into non-overlapping segments of length Δt and calculated the average decision value for the current context across all 100-ms bins within this segment. If the mean context decision fell closer to the correct context, that value for that individual segment was 0, otherwise it was 1. We then averaged over all segments of length Δt to obtain the mean accuracy of context decoding for segments of this size. This procedure was repeated for 25 different random ensembles in each individual recording, leading to an average accuracy curve (Fig. 4f). The time to 90% decoding accuracy (using a fixed sample size of 50 randomly drawn cells) was then determined as the first Δt for which the mean context-decoding accuracy exceeded 90% (Fig. 4g).

### Decoding using spatial versus mean-rate templates

To determine whether context-specific spatial maps carried contextual information, we either constructed location- and context-stratified decoding templates (Supplementary Fig. 8b, left) or a simplified context-only template where solely the mean activity rates across the entire linear track (familiar or novel, respectively) were used to construct population vectors for the decoding template (Supplementary Fig. 8b, right). We then compared decoding performance on the test runs for each of these templates per dataset by building a ratio between the two obtained error values. A ratio below 1 would indicate that the decoding performance of the template using spatial and contextual information was superior to one that used context-dependent firing rate differences only (Supplementary Fig. 8c).

### Statistics

The reported *n* indicate numbers of cells and exclude missing ("NaN") values. Unless otherwise stated, error bars are showing standard errors of the mean. On whisker plots, boxes are showing 25th to 75th percentiles, while whiskers indicate the 99% range. All statistical tests are described in the corresponding figure legends. Unless otherwise stated, statistical comparisons were made between cells fulfilling the specific criteria as indicated in the figure legends. To allow the use of ANOVAs on data that initially do not fulfill all the required assumptions (e.g., normally distributed population and/or common variance), we used a recently-developed tool named ARTool (available at https://depts.washington.edu/acelab/proj/art/) created by Wobbrock et al.[95,96]. In brief, this approach offers to apply an additional align-and-rank procedure to the raw data before proceeding with regular ANOVA. This preliminary step ensures that the resulting ANOVA will have main effects and interactions with appropriate Type I error rates and suitable power. Furthermore, in their most recent work, the authors extended the ART approach with an additional procedure referred to as ART-C[95] developed to facilitate post hoc pairwise comparisons. In the present paper, we used this tool to analyze data such as correlation values (e.g., Fig. 3c–h and Supplementary Fig. 5c–h). When this approach was not necessary, we used two- or three-way repeated measures ANOVAs (e.g., Fig. 4). In all cases, post hoc pairwise comparisons were performed using Tukey's test. All comparisons were two-sided, and the null hypothesis was rejected at the $P < 0.05$ level.

### Reporting summary

Further information on research design is available in the Nature Research Reporting Summary linked to this article.

## Data availability

Original/raw data reported in this study are available from the lead corresponding authors upon reasonable request. Any additional information required to reanalyze the data reported in this paper is available from the lead contact upon reasonable request. Source data are provided with this paper.

## Code availability

The original code of our Two-photon calcium imaging pipeline (MATLAB code) has been deposited at Zenodo.org (https://doi.org/10.5281/zenodo.5410473) and is publicly available as of the date of publication.

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

## Acknowledgements

We thank Dr. T. Hainmueller and Dr. J.-F. Sauer for reading and discussing earlier versions of the manuscript. We thank K. Winterhalter and K. Semmler for technical support and Dr. A. Kilias for her analytical support. This work was funded by the German Research Foundation (DFG BA1582/12-1 M.B.; FOR2143 M.B.), BMBF DLR 01GQ1901 DG-GC (M.B.) and by the ERC-AdG 787450 (M.B.).

## Author contributions

T.C. and M.B. designed the experiments, conceived the study, and wrote the manuscript. T.C. recorded and analyzed all imaging data, and performed the decoding analyses.

## Funding

## Competing interests

The authors declare no competing interests.
