## [Peer Review File · Nature Communications]

Hemisphere-specific spatial representation by hippocampal granule cellsReviewers' Comments:

Reviewer #1:

Remarks to the Author:

In this work, Cholvin and Bartos compared spatial coding properties of dentate gyrus granule cells (GCs) across brain hemispheres in mice running head-fixed. Using 2-photon calcium imaging, they were able to record large populations of GCs while mice run in familiar or novel virtual linear tracks across several days. The main finding reported is that both, spatial coding of the same track and discrimination of the two different contexts by GCs, were higher in the left than the right hemisphere. Similar differences in spatial, contextual coding and also the plasticity of memory representations have been previously reported for the CA1 and CA3 regions of mice, but this is the first time an across hemispheres comparison of this kind has been performed for the dentate gyrus. The experimental and analytical methods used are sound, the data is clearly presented and strongly supports authors' claims and interpretations, and the manuscript is well written. I did not find major concerns with the work presented so I list below a series of minor points aimed at improving clarity or offering additional support for some claims. I believe this manuscript would be of interest for the broad audience of Nature Communications as it fills a gap in the current studies of inter-hemispheric differences across hippocampal subregions in rodents.

- N of animals, sessions and cells that went into each analysis are not well reported throughout the paper. This information should be added in all cases to either figure legends or corresponding section in the main text. As an example, I could not find the n of mice recorded in either Figure 1 legend or the first section of the results.
- I would caution about the use of the word 'engram'. That term has specific connotations (e.g. Josselyn and Tonegawa, Science, 2020). Through the text it seems that it is rather used to refer to ensembles of GCs place cells.
- Spatial information in Figure 2D,E was quantified as bits per second. While this is a valid a common metric, it is also influenced by activity rate of the cells. Since these are already different between the left and right hemisphere, it may confound the analysis. A better metric would be to quantify spatial information in bits per spike (Skaggs et al., 1996).
- The section of the results describing Figure 3A-E is a bit confusing because multiple different analyses are mentioned but not explained (they are only described in the Methods). In particular, it should be clarified when an analysis was done comparing activity rates versus spatial map correlations, and the different indices (e.g. reliability, consistency) defined in the text.
- The Discussion section is clear and well balanced, but I missed some specific discussion about the mechanisms that could account for the underlying differences in spatial and context coding across hemispheres. There is extensive bibliography on this topic that would be of relevance here. A non-exhaustive list of potential mechanisms includes, entorhinal inputs (Fernandez-Ruiz et al., Science, 2021), 'backprojection' from CA3 (Scharfman, ProgBrainRes, 2007), local inhibitory inputs (Savanthrapadian, JNeurosci, 2014), etc.

Reviewer #2:

Remarks to the Author:

In this study, Cholvin & Bartos uses 2-photons calcium imaging and an alternance of familiar and novel virtual tracks to investigate inter-hemispheric differences of the dentate gyrus spatial code. Place cells show on average higher spatial tuning, spatial reliability and context discrimination in the left than in the right DG. Accordingly, place and context are better decoded by population activity of the left DG.

Overall the data and analyses are straightforward and of high quality, and the manuscript is well written. The authors made interesting links with previous reports on hemispheric specialization. I have only a few comments that would need addressing.

Specific comments:

Results

1) Animals ran faster on average in the novel environment (Fig.1H). Is it simply because the number of rewarded positions is smaller in the novel than in the familiar environment? How running speeds compare for similar segments like in between the reward positions 2 and 3?

2) Running speed for the left hemisphere group seems consistently lower (Fig.1H). Even though it is not a significant difference, this raise some uncertainty regarding the quantitative differences measured between left and right DG spatial codes. The authors need to show that the results (Fig2A and D) still hold for a selection of mice groups with similar running speed (maybe by excluding some outlier trials/mice).

3) Did the area monitored in left and right hemispheres had equivalent locations along the lateral and septo-temporal axes? I wonder if the authors could also tell, through the match of 2-photon and confocal images, whether similar radial levels were monitored.

Discussion

4) "These data may reconcile current inconsistencies in the literature reporting different degrees of spatial discrimination on the level of the DG network (Treves & Rolls, 1994; Knierim & Neunuebel, 2016; Allegra et al., 2020; Senzai & Buzsáki, 2017; Hainmueller & Bartos, 2018; Cholvin et al., 2021)."

_ Is there a correlation between remapping strength and recorded hemisphere across studies? Another factor likely important to explain these inconsistencies is the different nature of the environments and environmental alterations used across studies.

5) "Consistent with the finding that optogenetic silencing of the left but not the right CA3 impaired the performance in a long-term spatial memory task, it appears that the two hippocampal hemispheres do not provide the same information to CA1."

_ A reference would be needed here

6) "Thus, we propose that weighted inputs of the two DGs may influence the switch between the recall a familiar or the formation of a novel representation."

_ It is not totally clear how this proposition derives from the sentences that precede and matches the current findings. In general, how the stronger and weaker spatial tuning in left/right DG might relate to their respective storage and recall functions?^{SEP}

7) Figures

_ Figure 4A: adding on the top some labels indicating trials and environments would facilitate understanding

_ FigureS1-right panels might not be very meaningful

8) The study of Kim, Jung and Royer 2020 (Nature Communications) would be a particularly relevant reference for statements regarding emergence of context representations over time with learning and DG discrimination of contexts.

Reviewer #3:

Remarks to the Author:

The authors have previously shown that granule cells (GCs) in the mouse dentate gyrus (DG) can both discriminate between and generalise across different contexts (Chauvin et al., Neuron 2021). Here they report that these functions are lateralized such that left GCs are biased towards discrimination

and right GCs are biased towards generalization. This suggests that there is an hemispheric asymmetry upstream of the well-studied left-right asymmetry observed in hippocampal CA3. I believe these findings will be of significant interest to a broad range of neuroscientists interested in learning and memory mechanisms and hippocampal function. However, I have some concerns that should be addressed before the paper is published.

Major points:

1. Reporting of statistics. It is not always clear what hypothesis is being tested with each statistical test. There are a high number of p values reported, and some may need to be corrected for multiple comparisons. The most important hypothesis is probably the comparison between left and right side. Ideally, comparing left and right DG should of course be done in the same animal, but I can see why this may be experimentally challenging. Nevertheless, when comparing recordings from the left and right DG in groups of animals, the statistical N should be animals, not each novel environment an animal is exposed to, as has been done in the paper. This is not statistically appropriate for two reasons. First, despite the authors stating that these exposures are independent, they are not, as the animals are the same (taking the authors argument to the extreme, one could then test only two animals, one left and one right, repeatedly). Second, the exposure to a second novel environment is not equivalent to exposure to the first novel environment because the animal will have learnt from the first novel exposure. Therefore, I believe the proper statistical comparison is between the exposure to the first novel environment in each animal imaged on the left or right side. Moreover, it is important that the authors report the statistical N used for statistical testing in each figure legend, and not only refer the reader to a supplemental spreadsheet (which appears to simply report the output from a statistics software package without further explanation).

2. Interpretation of results. Whilst it is easy to agree that GCs in the left DG discriminate better between different contexts than GCs in the right DG do, the concept of generalization is not as straightforward. In my book, the term generalization would suggest that the cells learn what is similar in two contexts rather than code similar features irrespective of spatial context to start with (which is probably what most neurons in the brain do, and we do not say their function is generalization). To claim that the cells generalize one would therefore need to know what these cells actually code for: are they conventional place cells, or do they encode time or distance? And second, one would need to know the temporal development of this coding. I could not see evidence in the data presented that the GCs in the right DG actually learn the similarity of features in the two contexts. If I understand Figure 3H correctly, the cells start out with higher correlation between familiar and novel context from Day 1, and do not develop higher correlation over days, suggesting that, from Day 1, they code for the same feature in the two contexts, and do not learn ('generalize') across the contexts.

Minor points:

1. Some of the figure panels seem disproportionately small compared to others, e.g. Figure 1F,G are difficult to read unless you magnify 200-300%, and the blue traces of the linear track position could use an expanded y axis to make it stand out more clearly from the black baseline calcium trace. The pie charts in Figure 2G,H and Figure 3I,J,K are also disproportionately small compared to the other panels. In general, it would be an advantage with more uniform font sizes for axis labels.

Details:

In the pie charts in Figure 2G,H it would appear more intuitive to place the 'Both' (possibly in alternating stripe pattern) between the 'Familiar'-only and 'Novel'-only, as the reader can then more easily compare the total Familiar and total Novel.

In figure 4B,C Context error (%); should probably not be per cent, which would go from 0 to 100 (not 0 to 1).

Page 6, line 144 and page 11, line 249: What do the authors mean by 'larger engrams'? Do they simply mean accumulation of place cells with context-dependent place fields?

Point-by-point response

We would like to thank the reviewers for highlighting the novelty of our work ‘..it fills a gap in the current studies of inter-hemispheric differences across hippocampal subregions in rodents...’, emphasising the novelty and importance of our study ‘...this is the first time an across hemispheres comparison of this kind has been performed for the dentate gyrus...’, ‘.....these findings will be of significant interest to a broad range of neuroscientists interested in learning and memory mechanisms and hippocampal function.....’ and its methodological high quality ‘...analyses are straightforward and of high quality...’. Nevertheless, some comments have been raised which we aim to address in the following.

Reviewer #1:

In this work, Cholvin and Bartos compared spatial coding properties of dentate gyrus granule cells (GCs) across brain hemispheres in mice running head-fixed. Using 2-photon calcium imaging, they were able to record large populations of GCs while mice run in familiar or novel virtual linear tracks across several days. The main finding reported is that both, spatial coding of the same track and discrimination of the two different contexts by GCs, were higher in the left than the right hemisphere. Similar differences in spatial, contextual coding and also the plasticity of memory representations have been previously reported for the CA1 and CA3 regions of mice, but this is the first time an across hemispheres comparison of this kind has been performed for the dentate gyrus. The experimental and analytical methods used are sound, the data is clearly presented and strongly supports authors’ claims and interpretations, and the manuscript is well written. I did not find major concerns with the work presented so I list below a series of minor points aimed at improving clarity or offering additional support for some claims. I believe this manuscript would be of interest for the broad audience of Nature Communications as it fills a gap in the current studies of inter-hemispheric differences across hippocampal subregions in rodents.

We are very grateful that the reviewer highlights the experimental challenges and the importance of our study for the scientific community.

Major comments:

1. N of animals, sessions and cells that went into each analysis are not well reported throughout the paper. This information should be added in all cases to either figure legends or corresponding section in the main text. As an example, I could not find the n of mice recorded in either Figure 1 legend or the first section of the results.

We thank the reviewer for pointing out this oversight. We modified the manuscript according to the reviewer’s request. Notably, we indicated the numbers of animals, datasets and cells that entered each analysis represented by the figures, either directly in the main text, in the legend or in the figures of the revised manuscript.

2. I would caution about the use of the word ‘engram’. That term has specific connotations (e.g. Josselyn and Tonegawa, Science, 2020). Through the text it seems that it is rather used to refer to ensembles of GCs place cells.

This point has also been raised by reviewer #3 and led us to reconsider the use of the word “engram”, which is now explicitly defined at the beginning of the manuscript (page 3, line 50-53). Indeed, this word has been replaced by the formulation suggested by the reviewer (ensembles of GC representing a context) at several occasions within the revised manuscript (e.g. page 4 line 86; page 5, line 131 and page 6, line 145).

3. Spatial information in Figure 2D,E was quantified as bits per second. While this is a valid a common metric, it is also influenced by activity rate of the cells. Since these are already different between the left

and right hemisphere, it may confound the analysis. A better metric would be to quantify spatial information in bits per spike (Skaggs et al., 1996).

We thank the reviewer for her/his suggestion on further data analysis. Here we would like to argue that bits/second is an appropriate measure for spatial information. Indeed, the metric bits/spike has been shown to be sensitive to differences in firing rate, and thus the calculation of SI based on bits/second has been employed in electrophysiological studies to verify the results obtained using bits/spike (e.g. Wirtshafter & Wilson, eLife, 2020). Moreover, while action potentials recorded with electrophysiological techniques constitute discrete temporal events, calcium signals obtained by 2-photon imaging are an indirect readout of cellular activity, preventing a direct computation of the metric bits/spike. To express calcium data in bits/spike, one would first have to rely on spike deconvolution algorithms, which infer a spike train under the assumption that the fluorescence trace represents an approximate convolution of the underlying spike train. It appears that accurately inferring spike times from calcium signals is a challenge, and different algorithms lead to different results (for a comparison, see e.g. Pachitariu et al., 2018, J Neuroscience). Thus, this approach, which requires extra-steps of data processing, is inevitably combined with the risk of not getting necessarily closer to the ground truth.

In the present study, we used the same method as in our previous work (Hainmueller et al. 2018, Nature; Cholvin et al. 2021, Neuron) for SI assessment. This method is adapted from Skaggs et al., (1993; Advances in Neural Information Processing Systems) for calcium signals, such that the calculation of the spatial information (SI) content of a cell is taking the activity level of the cell into account. More precisely, SI content is inferred from differences between the activity in a specific bin (location) of the track and the overall calcium activity averaged over the entire track (see also Methods, section 'Activity differences and spatial information', for more details, including formula; page 19, lines 491-506). Thus, we are convinced that the metric we used for SI assessment is appropriate and adapted to the limitations inherent to calcium imaging, and hope that our explanation is convincing for the reviewer.

4. The section of the results describing Figure 3A-E is a bit confusing because multiple different analyses are mentioned but not explained (they are only described in the Methods). In particular, it should be clarified when an analysis was done comparing activity rates versus spatial map correlations, and the different indices (e.g. reliability, consistency) defined in the text.

*We thank the reviewer for pointing out this element of potential confusion. Each term is now directly defined in the figure legends and in the results section on **page 6, line 160** for reliability, and on **page 7, line 168** for consistency. The description on the applied statistical analysis was improved in the results section and in the corresponding figure legends of the revised manuscript.*

5. The Discussion section is clear and well balanced, but I missed some specific discussion about the mechanisms that could account for the underlying differences in spatial and context coding across hemispheres. There is extensive bibliography on this topic that would be of relevance here. A non-exhaustive list of potential mechanisms includes, entorhinal inputs (Fernandez-Ruiz et al., Science, 2021), 'backprojection' from CA3 (Scharfman, ProgBrainRes, 2007), local inhibitory inputs (Savanthrapadian, JNeurosci, 2014), etc.

*We thank the reviewer for her/his proposal on how to improve the discussion. We added a new paragraph dedicated to the potential mechanisms that could account for the observed differences in spatial and context encoding across hemispheres of the revised manuscript on **page 11 starting at line 293**, in which we discuss the potential contribution of entorhinal cortex glutamatergic inputs, lateral inhibition by local GABAergic cells and inputs by back-projecting CA3 principal cells in the functional lateralization of DG place cells.*

Reviewer #2:

In this study, Cholvin & Bartos uses 2-photon calcium imaging and an alternance of familiar and novel virtual tracks to investigate inter-hemispheric differences of the dentate gyrus spatial code. Place cells show on average higher spatial tuning, spatial reliability and context discrimination in the left than in the right DG. Accordingly, place and context are better decoded by population activity of the left DG.

Overall the data and analyses are straightforward and of high quality, and the manuscript is well written. The authors made interesting links with previous reports on hemispheric specialization. I have only a few comments that would need addressing.

We would like to thank the reviewer for her/his positive feedback praising our work, the quality of our data and the well-written manuscript.

Major comments:

1. Animals ran faster on average in the novel environment (Fig.1H). Is it simply because the number of rewarded positions is smaller in the novel than in the familiar environment? How running speeds compare for similar segments like in between the reward positions 2 and 3?

*We are grateful for the opportunity to clarify our data. Following the reviewers request, we plotted in **Figure 1** for reviewers (see below) the running speeds for the segment between the reward positions 2 and 3 (left plot) and compared them with the original data for the running speed for the entire track (right plot).*

Indeed, both plots appear to be very similar and fit to our experience that the number of reward sites has no significant effect on running speed. This could be partially explained by the fact that in this analysis, we considered only running periods during which mice ran with a speed of at least $5 \text{ cm}\cdot\text{s}^{-1}$ (see also Hainmueller et al., Nature 2018; Cholvin et al., Neuron 2021), as this is also the threshold we used when computing calcium data. Thus, if animals stopped at reward sites and took their time to lick, which sometimes happens, these immobility periods would not be considered in our analyses. However, we noticed that the running speed in novel environments are always mildly higher compared to familiar ones, independent of the number of reward sites. This observation is based on several projects running in our laboratory. We therefore tend to attribute this to the novelty factor associated with new virtual environments.

2. Running speed for the left hemisphere group seems consistently lower (Fig.1H). Even though it is not a significant difference, this raise some uncertainty regarding the quantitative differences measured between left and right DG spatial codes. The authors need to show that the results (Fig2A and D) still hold for a selection of mice groups with similar running speed (maybe by excluding some outlier trials/mice).

We thank the reviewer for bringing up this issue. Indeed, and as noted by the reviewer, the running speed was not significantly different between left and right DGs of imaged animals ($p = 0.143$). Nevertheless, we followed the reviewer's request and examined how the results would look after exclusion of outliers by removing 2 datasets from the 18 datasets per group (left and right DGs). Indeed, removing those few datasets improved the similarity in the mean running speeds of left- and right- implanted mice (**Figure 2** for reviewers; see below).

Following the request of the reviewer, we replicated the analyses shown in the original Figure 2 A/D for these selected datasets (**Figure 3** for reviewers; see below; left new analysis, right original analysis).

In this new Figure, we show that the removal of the two datasets had a minor effect on the mean activity and spatial information of cells in both hemispheres. Moreover, the spatial information content

remained significantly different between the left and right DGs, confirming our initial conclusions. We hope that these new analyses provided here remove any uncertainty on our results and their interpretation.

3) Did the area monitored in left and right hemispheres had equivalent locations along the lateral and septo-temporal axes? I wonder if the authors could also tell, through the match of 2-photon and confocal images, whether similar radial levels were monitored.

Thank you for this question. **First**, we always imaged the upper blade, in a central position along the medio-lateral axis, which corresponds to coordinates $AP = 2.0$ mm, $ML = 1.2$ mm, $DV = 1.8$ mm from brain surface and $600-700$ μm below the window (as illustrated by the red dotted line in the original Figure 1B). **Second**, this area is the most dorsal (i.e. the closest to the window surface) and, thus, offers the best imaging quality (especially when considering that we image the DG without removing CA1). Below in **Figure 4** for reviewers we show an example illustrating how we typically define 2 fields of view recorded in an animal. These coordinates and settings have been applied to both hemispheres, i.e. similar locations along the lateral and septo-temporal axes were used for imaging population activity in the left and right DGs.

Third, due to the dome-like shape of the granule cell layer in the imaged areas, recorded fields of view always comprise cells situated superficially, in the middle and in the deep granule cell layer. All imaged cells were located in the same field of view and optical plane under the microscope. Thus, each dataset includes granule cells located all along the radial axis in both hemispheres.

*Taken together, we used a total of 18 datasets per group (left/right), recorded similar areas at similar depth and medio-lateral coordinates in all animals, while encompassing the entire radial axis of the granule cell layer. We are thus confident that our data are based on populations of granule cells comparable between left and right DGs. Thanks to the reviewer's request, information on the precise location of the imaging regions is now included in the Methods section on **page 18, line 472-477** of the revised manuscript.*

4. "These data may reconcile current inconsistencies in the literature reporting different degrees of spatial discrimination on the level of the DG network (Treves & Rolls, 1994; Knierim & Neunuebel, 2016; Allegra et al., 2020; Senzai & Buzsáki, 2017; Hainmueller & Bartos, 2018; Cholvin et al., 2021)." Is there a correlation between remapping strength and recorded hemisphere across studies? Another factor likely important to explain these inconsistencies is the different nature of the environments and environmental alterations used across studies.

The reviewer highlights an important point regarding the comparison of results across studies. Unfortunately, most studies do not indicate which hemisphere has been imaged or electrophysiologically recorded; thus, making comparisons among studies is basically impossible. To the best of our knowledge, the decision of which hemisphere to record from is often not taken based on scientific considerations, but driven by more practical aspects (e.g., experimental setup limitations or the experimenter being left- or right-handed and being more comfortable performing surgeries on one side, etc.). We hope that our study can contribute to a better consideration of this variable, starting with the simple act of reporting which hemisphere(s) has been studied.

*We fully agree with the reviewer that different environments and environmental alterations are likely to play a role in the aforementioned inconsistencies across studies. We did not pretend to explain all of the discrepancies that exist in the literature about DG function based on our results, but rather wished to highlight that lateralization is an important factor, and that so far it appears that this factor has not been appreciated. Consequently, we toned down our statement to 'These data may help to reconcile current inconsistencies in the literature reporting different degrees of spatial discrimination on the level of the DG network (Treves & Rolls, 1994; Knierim & Neunuebel, 2016; Allegra et al., 2020; Senzai & Buzsáki, 2017; Hainmueller & Bartos, 2018; Cholvin et al., 2021).' We also added another sentence in accordance with the reviewer's remark on **page 10, lines 259-263**: 'However, other factors such as different settings of virtual realities and different degree of their alterations may also play a role.'*

5. "Consistent with the finding that optogenetic silencing of the left but not the right CA3 impaired the performance in a long-term spatial memory task, it appears that the two hippocampal hemispheres do not provide the same information to CA1." A reference would be needed here

We thank the reviewer for noticing the omission of a citation and refer to Shipton et al. (2014) in the revised version of the manuscript.

6. "Thus, we propose that weighted inputs of the two DGs may influence the switch between the recall a familiar or the formation of a novel representation." It is not totally clear how this proposition derives from the sentences that precede and matches the current findings. In general, how the stronger and weaker spatial tuning in left/right DG might relate to their respective storage and recall functions?

We agree that this sentence lacks clarity. We therefore modified the sentence to: 'Thus, we propose that inputs of the two DGs may influence the switch between the recall of an existing memory and the formation of a novel representation in downstream hippocampal areas'. The rationale behind this proposition is that just like the modulatory signals originating from parahippocampal structures and inputs from the entorhinal cortex (EC), which are providing information about the saliency of novel environments, whether a familiar

(and thus already stored) representation is updated, or a novel representation is formed, may also depend on the input provided by the two DGs. As we observed a bias towards context-specific representations in the left DG, and a bias towards generalization of shared properties in the right DG, we hypothesize that in a case of uncertainty (as a spatial context that would be just slightly different from a previously-explored one, e.g. by partially modifying a familiar environment), the balance between these two left and right sets of GCs involved in the encoding of the known environment may play a role in the network's tendency to recall an existing memory or engage in the formation of a novel representation.

7. Figure 4A: adding on the top some labels indicating trials and environments would facilitate understanding

We thank the reviewer for this advice. We modified the figure accordingly.

8. FigureS1-right panels might not be very meaningful

In accordance with the reviewer's request, we removed the right panels from the Figure S1B and C.

9. The study of Kim, Jung and Royer 2020 (Nature Communications) would be a particularly relevant reference for statements regarding emergence of context representations over time with learning and DG discrimination of contexts.

Many thanks for highlighting this elegant study, which is relevant to our hypotheses regarding the emergence of context representations and the reasoning we follow in our manuscript. It is now cited in the revised manuscript.

Reviewer #3:

The authors have previously shown that granule cells (GCs) in the mouse dentate gyrus (DG) can both discriminate between and generalize across different contexts (Chauvin et al., Neuron 2021). Here they report that these functions are lateralized such that left GCs are biased towards discrimination and right GCs are biased towards generalization. This suggests that there is a hemispheric asymmetry upstream of the well-studied left-right asymmetry observed in hippocampal CA3. I believe these findings will be of significant interest to a broad range of neuroscientists interested in learning and memory mechanisms and hippocampal function. However, I have some concerns that should be addressed before the paper is published.

We would like to thank the reviewer for highlighting the relevance of our study for the broader scientific community interested in learning and memory.

Major comments:

1. Reporting of statistics. It is not always clear what hypothesis is being tested with each statistical test. There are a high number of p values reported, and some may need to be corrected for multiple comparisons.

We are grateful for the opportunity to clarify our statistical analysis. We aimed at testing the hypotheses raised by the experimental design by using either 2- or 3-way ANOVA tests. This approach allows to test for main effects of 2 or 3 variables that could explain the results (left vs right DG; familiar vs novel context; across days). For example, Figure 2D shows the "Mean spatial information of active (>1 transient/min) GCs either from left- or right-implanted mice recorded over 5 consecutive days exposed to familiar and novel environments." In this case, 3 parameters could influence the results: the side of the brain (left / right), the context (familiar / novel) and the day of recording (1 to 5). A 3-way ANOVA tests for these hypotheses all at once. We provide the results of the statistical tests (main effects) directly on the figure under the labels

“Side”, “Context” and “Days”. Then, in the eventuality of the presence of a significant effect (e.g. in Figure 2D, the variables “Side” and “Days” are showing significant differences), we used the classical approach consisting of a multiple comparison procedure to isolate which group(s) differ from the others (Tukey’s post-hoc test; e.g. testing between left and right imaged animals at day 1 in one environment). We fully agree that the explanation on the terminologies ‘Days; Side and Context’ should be better specified and included an explanation in the Figure 2 legend of the revised manuscript.

Furthermore, as highlighted in the Method section of the manuscript, we used a recently-developed tool named ARTool (available at <https://depts.washington.edu/acelab/proj/art/>) created by Wobbrock et al. (Wobbrock et al., 2011; Elkin et al., 2021), designed to allow the use of ANOVAs on data that initially do not fulfill all the required assumptions (i.e. normally distributed population and / or common variance). In brief, this approach offers to apply an additional align-and-rank procedure to the raw data before proceeding with regular ANOVA tests. This preliminary step ensures that the resulting ANOVA will have main effects and interactions with appropriate Type I error rates and suitable power. Finally, regarding the high number of *p* values reported, we simply wanted to be as detailed and transparent as possible. At the same time, we did not aim to overload the manuscript with *p* values, and therefore reported them in the Table S1.

The most important hypothesis is probably the comparison between left and right side. Ideally, comparing left and right DG should of course be done in the same animal, but I can see why this may be experimentally challenging. Nevertheless, when comparing recordings from the left and right DG in groups of animals, the statistical N should be animals, not each novel environment an animal is exposed to, as has been done in the paper. This is not statistically appropriate for two reasons. First, despite the authors stating that these exposures are independent, they are not, as the animals are the same (taking the authors argument to the extreme, one could then test only two animals, one left and one right, repeatedly). Second, the exposure to a second novel environment is not equivalent to exposure to the first novel environment because the animal will have learnt from the first novel exposure. Therefore, I believe the proper statistical comparison is between the exposure to the first novel environment in each animal imaged on the left or right side.

We would like to thank the reviewer for raising this important topic. **First**, we would like to highlight that in both left- and right-implanted animals, half of the animals (6 left- mice, 5 right- mice) were first exposed (dataset 1) to the environment identified as “Novel 1” (in Figure S1), then to “Novel 2” for replication (second dataset), while the other half of the animals (5 left- mice, 5 right- mice) were first exposed to “Novel 2” (dataset 1) and then to “Novel 1” (dataset 2). We designed the experiments this way, because we aimed to limit any potential influence of the order of exposure to the two novel environments on our data. **Second**, we would like to emphasize that among the 18 datasets we recorded from the left-implanted animals and the 18 datasets obtained from the right-implanted animals, 11 (left hemisphere) and 10 (right hemisphere) datasets were first-novelty exposure datasets, while the additional 7 (left hemisphere) and 8 (right hemisphere) datasets were second-novelty exposure datasets. These data went into the original Figure 2. Moreover, the number of datasets of each type is comparable between left- and right-implanted mice. Thus, if the exposure to a second novel environment had any effect on our results, one can expect that this influence should be present in both left- and right-implanted animals. **Third**, we followed the reviewer’s request and replicated the main analyses shown in the original Figure 2A, C, D, F and Figure 3C-H, by taking into account only the first-novelty exposure datasets for each animal. Thus, we re-analyzed activity difference scores in the left and right DG, place field reliability and session consistency in the left and right DG across days of mice exposed to the familiar and the initial novel environment as well as spatial correlations of place fields across days in the left and right DG of mice exposed to a familiar and one initial novel environment. These new data are shown in the new **Figure S7** and described on **page 8, line 198**. This new analysis reproduced our main original results and thereby strengthen their interpretation. This important verification has now been added to the supplemental material of the revised manuscript.

As a side note, we would like to emphasize that the number of imaged active GCs within one animal is very low due to the sparse activity within the DG. In order to keep the number of animals moderate and at the same time obtain a sufficient number of cells, we exposed mice to two novel environments. Finally,

our results are in agreement with the latest work of Priestly et al. (Neuron 2022) which exposed mice to a second novel environment during 2-photon imaging of CA1 place cells, and showed that they could repeat the results obtained during first novelty exposure.

Moreover, it is important that the authors report the statistical N used for statistical testing in each figure legend, and not only refer the reader to a supplemental spreadsheet (which appears to simply report the output from a statistics software package without further explanation).

We thank the reviewer for her/his suggestion and accordingly included this information in the figure legends.

2. Interpretation of results. Whilst it is easy to agree that GCs in the left DG discriminate better between different contexts than GCs in the right DG do, the concept of generalization is not as straightforward. In my book, the term generalization would suggest that the cells learn what is similar in two contexts rather than code similar features irrespective of spatial context to start with (which is probably what most neurons in the brain do, and we do not say their function is generalization). To claim that the cells generalize one would therefore need to know what these cells actually code for: are they conventional place cells, or do they encode time or distance? And second, one would need to know the temporal development of this coding. I could not see evidence in the data presented that the GCs in the right DG actually learn the similarity of features in the two contexts. If I understand Figure 3H correctly, the cells start out with higher correlation between familiar and novel context from Day 1, and do not develop higher correlation over days, suggesting that, from Day 1, they code for the same feature in the two contexts, and do not learn ('generalize') across the contexts.

*We would like to thank the reviewer for raising this interesting question which shows how important the exact definition of each term is for precise scientific discussions. In the present study (as well as in Hainmueller & Bartos, Nature 2018; Cholvin et al., Neuron 2021), discrimination and generalization define the two extremes of place field correlations between two distinct contexts (A and B) ranging from 0 (maximal discrimination) to 1 (maximal generalization between the two contexts – lack of discrimination). While we agree with the reviewer that mice can be trained to generalize between environments (e.g. Nakashiba et al., Neuron 2012), in our manuscript, generalization does not necessarily involve a specific temporal development during learning. It might also be noteworthy to mention that only the cells identified as place cells were considered when assessing discrimination / generalization (e.g. Figure 3). Indeed, as the reviewer noted, “the cells start out with higher correlation between familiar and novel context from day 1, and do not develop higher correlation over days, suggesting that, from day 1, they code for the same feature in the two contexts”, and this is what we define as generalization in the spatial representations of the two environments. We therefore included the terminology ‘instantaneous generalization between environments’ in the revised manuscript on **page 7, line 184**.*

3. Some of the figure panels seem disproportionately small compared to others, e.g. Figure 1F,G are difficult to read unless you magnify 200-300%, and the blue traces of the linear track position could use an expanded y axis to make it stand out more clearly from the black baseline calcium trace. The pie charts in Figure 2G,H and Figure 3I,J,K are also disproportionately small compared to the other panels. In general, it would be an advantage with more uniform font sizes for axis labels.

We modified the figures according to the reviewer’s suggestion.

4. In the pie charts in Figure 2G,H it would appear more intuitive to place the ‘Both’ (possibly in alternating stripe pattern) between the ‘Familiar’-only and ‘Novel’-only, as the reader can then more easily compare the total Familiar and total Novel.

In figure 4B,C Context error (%); should probably not be per cent, which would go from 0 to 100 (not 0 to 1).

We modified the figures accordingly.

5. Page 6, line 144 and page 11, line 249: What do the authors mean by 'larger engrams'? Do they simply mean accumulation of place cells with context-dependent place fields?

*A related question was also raised by reviewer #1 (see her/his point 2). We therefore defined the word 'engram' in the revised manuscript on **page 3, line 51** and now state throughout the manuscript that we examine ensembles of GCs. Indeed, the sentences that attracted the reviewer's attention were modified in the revised manuscript to 'However, this effect was more pronounced in the left hemisphere, resulting in larger ensembles of GCs place cells with improved spatial encoding properties in the left than right DG' (**page 6, lines 144-146**) and 'Finally, cell associations representing novel contexts increased across learning and resulted in larger GC place cell ensembles in the left DG, suggesting bilateral differences in the sparsity of representations' (**page 10, lines 254-256**).*

Reviewers' Comments:

Reviewer #1:

Remarks to the Author:

The authors have addressed all my initial comments and the current version of the manuscript has been significantly improved. I believe it is acceptable for publication in its present form and will be an important contribution to the field.

Reviewer #2:

Remarks to the Author:

The authors have well addressed all my concerns and I have no further comments.

Reviewer #3:

Remarks to the Author:

The authors have adequately addressed all my previous concerns, and, apart from the correction of some typos in the newly added sections, I believe this manuscript is ready for publication.